# Intramammary rapamycin administration to calves induces epithelial stem cell self-renewal and latent cell proliferation and milk protein expression

**Anna Kosenko[1,2], Shamay Jacoby[1], Tomer-Meir Salame[3], Maya Ross[1], Itamar Barash[1] ***

**1** Institute of Animal Science, Agricultural Research Organization, The Volcani Center, Bet Dagan, Israel,
**2** The Robert H. Smith Faculty of Agriculture, Food and Environment, The Hebrew University of Jerusalem, Jerusalem, Israel, **3** Life Sciences Core Facilities, Weizmann Institute of Science, Rehovot, Israel

* itamar.barash@mail.huji.ac.il

**Data Availability Statement:** All relevant data are within the paper and its Supporting information files.

## Abstract

Mammary epithelial stem cells differentiate to create the basal and luminal layers of the gland. Inducing the number of differentiating bovine mammary stem cells may provide compensating populations for the milk-producing cells that die during lactation. Inhibition of mTOR activity by rapamycin signals self-renewal of intestinal stem cells, with similar consequences in the mouse mammary gland and in bovine mammary implants maintained in mice. The implementation of these results in farm animals for better mammary development and production was studied in 3-month-old calves. mTOR activity decreased by ~50% in mammary epithelial cells subjected to 3-week rapamycin administration, with no negative consequences on mammary morphology or β-casein expression. Subsequently, stem cell self-renewal was induced, reflected by a higher propagation rate of cultures from rapamycin-treated glands compared to respective controls and higher expression of selected markers. Followed by 4-day estrogen and progesterone administration, rapamycin significantly induced proliferation rate. Higher numbers of basal and luminal PCNA$^+$ cells were detected in small ducts near the elongating sites as compared to large ducts, in which only luminal cells were affected. Rapamycin administration resulted in induction of individual milk protein genes' expression, which was negatively correlated to their endogenous levels. The inductive effect of rapamycin on luminal cell number was confirmed in organoid cultures, but milk protein expression decreased, probably due to lack of oscillation in rapamycin levels. In conclusion, intramammary rapamycin administration is an effective methodology to reduce mTOR activity in bovine mammary epithelial cells and consequently, induce stem cell self-renewal. The latent positive effect of rapamycin on epithelial cell proliferation and its potential to improve milk protein expression in calves may have beneficial implications for mature cows.

**Funding:** This study was supported by the Chief Scientist of the Ministry of Agriculture and Rural Development (Grant number 20-04-0060) and the Israeli Milk Marketing Board (Grant number 362-0574). The funders had no role in study design, data collection and analysis, decision to publish, or preparation of the manuscript.

**Competing interests:** The authors have declared that no competing interests exist.

## Introduction

Two major mammary characteristics: milk secretion and susceptibility to breast cancer, have paved the way for separate research avenues into improved farm animal production and human health, respectively. Interestingly, both involve a common effector—the mammary epithelial stem cell at the top of the mammary cell hierarchy. Despite a lack of definite morphological identification and debated multipotency of the entire stem cell population in the postpartum mouse mammary gland [1–4], which has not been confirmed in the human breast [5], these cells undoubtedly give rise to the basal and luminal layers that compose the functional mammary epithelium. They also renew the mammary gland between consecutive lactation periods.

Notable progress has been achieved in mouse mammary stem cell characterization since breakthrough studies were published, demonstrating reconstitution of a functional mammary gland by transplantation of a single or several stem cells into the de-epithelized mammary stroma [6, 7]. Better separation from progenitors has been performed [8], new effectors, markers and subpopulations have been identified [Reviewed by [4, 9]] and most importantly, unipotent luminal and basal stem cell populations with self-renewing potency have been identified in the postpartum mammary gland [1]. Together with their bipotent counterparts, these cells maintain the morphogenesis and homeostasis of the adult gland [2].

In contrast, only modest progress has been achieved in bovine mammary stem cell studies. This understudied population has been enriched by flow cytometry [10] and the cells' differentiation status has been followed with identified markers [11], but not by lineage tracing. It is only very recently that these stem cells' ability to generate a representative bovine mammary morphology has been demonstrated in bovine stroma pre-transplanted into the mammary stroma of immunocompromised mice [12].

Focusing on improved bovine milk synthesis, it has been suggested that the mammary gland's ability to replace the differentiated milk-producing luminal cells that die during lactation is key to maintaining persistent milk production [13]. Accordingly, a higher number of mammary epithelial stem cells that continuously differentiate into milk-producing luminal cells and their basal myoepithelial counterparts may provide a replacement for the apoptizing milk-producing cells, thus moderating the decrease in milk production.

An initial attempt was made to induce bovine mammary epithelial stem cell number in 3-month-old calves by intramammary xanthosine infusion for 5 consecutive days [14]. An elevated number of retained BrdU-labeled epithelial cells and induced telomerase activity were noted. Later, induced expression of stem cell gene markers by xanthosine was also observed in goats [15]. Nevertheless, xanthosine's self-renewing effect could not be reproduced in stem cells of transplanted bovine parenchyma implants, which were analyzed for BrdU-cell retention and by flow cytometry [16]. Moreover, xanthosine administration suppressed bovine mammary cell proliferation by 50%, due to inhibition of inosine-5′-monophosphate dehydrogenase activity, a rate-limiting enzyme in guanine synthesis.

Calorie restriction (CR) induced self-renewal of intestinal mouse stem cells [17]. These cells do not directly sense the reduced caloric intake [18]; the respective self-renewing signal is delivered by paracrine cADP ribose secreted from neighboring Paneth cells that are subjected to decreased mammalian/mechanistic target of rapamycin (mTOR) activity. Pharmacological inhibition of mTOR activity by daily rapamycin injections reproduced the inducing effect of CR on intestinal stem cell self-renewal [17]. Interestingly, oral administration of rapamycin by gavage blocked this CR effect, probably due to endogenous inhibition of mTOR activity in the intestinal stem cells [18].

Pursuing a reliable approach for inducing stem cell number in the mammary gland of farm animals, the aforementioned studies were initially reproduced in intact mice and those carrying bovine mammary implants in their mammary stroma that were subjected to CR for 6–8 weeks [19]. In both models, CR induced a shift from mammary stem cell differentiation to self-renewal, which was reproduced by a shorter, 3-week period of rapamycin administration. Moreover, latent effects of rapamycin administration on improved mammary development and production were noted.

The model-based positive effect of rapamycin on mammary epithelial stem cell self-renewal was further investigated here in bovine. It was hypothesized that skip-a-day intramammary rapamycin administration at a defined concentration will cause oscillating mTOR activity in mammary parenchymal cells. During intervals of low mTOR activity, paracrine signaling for stem cell self-renewal will ensue from an as-yet undefined niche population. Higher mTOR activity will enable internal self-renewal of the stem cells. It was anticipated that increasing the epithelial stem cell reservoir will latently improve mammary activities associated with development and production.

We report that rapamycin, administered via the nipple into the mammary ductal system of 3-month-old calves, penetrates the lining epithelial structures and suppresses S6 phosphorylation with no effect on β-casein synthesis or mammary morphology. Subsequently, enhanced stem cell self-renewal was reflected by the longer propagation of cultures prepared from rapamycin-treated glands compared to their respective controls. Latent rapamycin effects included induced luminal and basal cell proliferation in a location- and cell type-dependent manner, as well as induced milk protein gene expression that was negatively correlated to the endogenous milk protein expression potential of the individual calf.

## Materials and methods

### Ethics statement

All animals used in this study were treated humanely. Study protocols were in compliance with the regulations of the Israeli Ministry of Health and the Animal Experimentation Ethics Committee of the Agricultural Research Organization, Volcani Center.

### Animals

Three-month-old female Holstein calves (~90 kg body weight [BW]) that were raised on the Volcani Center's experimental farm were used in these experiments. During the experimental periods, each mammary gland was administered via the teat canal with 2 mL of vehicle, composed of sterile PBS containing 10% polyethylene glycol (Sigma, St. Louis, MO) and 10% Tween 80 (Sigma). In individual experiments, the vehicle was supplemented with Evans blue to analyze the penetration of this solution into the mammary epithelium, or with rapamycin to analyze its effect on cellular activities. The vehicle was administered using a sterile 22G IV cannula (Deltaven, Viadana, (MN) Italy), cut to ~1.5 cm and inserted (without the needle) into the teat canal. This procedure did not involve any sedation or anesthesia. In the relevant experiments, estrogen (0.1 mg/kg per day, Sigma) and progesterone (0.25 mg/kg per day, Sigma) were administered following the rapamycin treatment by daily subcutaneous injections for 4 consecutive days [20]. The design of the studies was based on the distinct morphology and activity of the individual "quarters" (i.e., glands) of the bovine udder [21], which can be individually affected [14]. Thus, in studies analyzing the rapamycin effects, front or rear glands of each side of the calf served as internal controls for the other two rapamycin-administered glands.

At the end of the experimental period, calves were heavily sedated with intravenous xylazine (20 mg/mL; 0.1 mg/kg, Sedaxylan Veterinary, Abic Phibro, Bet Shemesh, Israel) and

ketamine (1 g/10 mL; 0.5 mg/kg, Clorketam, Vetoquinol, Lure, France) and sacrificed with intravenous 20% embutramide, 5% mebezonium iodide and 0.5% tetracaine hydrochloride (T-61; 0.15 mL/kg, Intervet Canada Corp., Quebec, Canada). The complete udder was then removed for further analyses.

## Morphological analysis by whole-mount examination

For whole-mount examination, bovine mammary parenchymal tissue pieces (~0.5 cm$^3$) were excised from the udder, fixed, and stained with Carmine red as previously described [22]. Stained whole mounts were visualized and photographed using a binocular equipped with cell-Sens standard 1.4 software (Olympus Scientific Solutions, Tokyo, Japan).

## Histological analysis, immunohistochemistry and immunofluorescence

Bovine mammary samples were fixed in Bouin's solution, dehydrated in a graded ethanol series (50% to 100%), cleared in xylene and embedded in paraffin. Paraffin sections (5 μm) were stained with hematoxylin and eosin (H&E; Sigma) to visualize the morphology of the mammary layers.

Immunohistochemistry was performed on the 5-μm paraffin-embedded sections after antigen retrieval. The reaction with primary antibodies was initiated by blocking the sections with 10% bovine serum albumen and 5% goat serum in 0.1% PBST for 1 h prior to overnight incubation with the first antibody at 4°C. Sections were washed and incubated with N-Histofine (Nichirei Biosciences, Tokyo, Japan) for 30 min at room temperature. Signals were generated with 3,3-diaminobenzidine (DAB) substrate (Vector Laboratories, Burlingame, CA). For immunofluorescence analyses, the sections were incubated with secondary antibodies for 1 h at room temperature. Nuclei were stained with 4',6-diamidino-2-phenylindole (DAPI) (Qbiogen, Irvine, CA). The antibodies and their dilutions are listed in S1 Table. In all of the analyses, stained tissues were visualized and photographed under an inverted fluorescence microscope (Eclipse Ti, Nikon Instruments, Melville, NY) equipped with NIS-Elements AR 3.2 imaging software (Nikon Instruments), or an Olympus IX 81 inverted laser scanning confocal microscope (FluoView 500, Tokyo, Japan).

## Flow cytometry

Bovine mammary tissue pieces were harvested from the parenchymal region of each gland, digested into organoids and kept at -80°C as previously described [10]. For flow cytometry, organoids were washed and further digested into dispersed mammary cells. Lin⁻ (CD45, TER119, CD31 and BP-1) cell suspension was prepared using the EasySep mouse epithelial enrichment kit according to the manufacturer's protocol (STEMCELL Technologies, Vancouver, Canada), labeled with PE-conjugated anti-CD24 and FITC-conjugated anti-CD49f antibodies, and sorted as previously described [10, 11, 23]. The list of antibodies is provided in S1 Table. Cell viability was tested by staining with BD Horizon Fixable Viability Stain 450 diluted 1:500 (BD Biosciences, San Jose, CA). Cell sorting was performed in a FACSAria III cell sorter (BD Biosciences) at the Department of Biological Services of the Weizmann Institute of Science (Rehovot, Israel).

## RNA extraction and real-time PCR analyses

RNA was extracted from the parenchymal region of each mammary gland using TRIzol reagent and reverse-transcribed with the qScript Synthesis Kit (Quanta BioSciences, Beverly, MA). RNA quality and quantity were determined in a NanoDrop 1000 spectrophotometer

(Thermo Fisher Scientific, Wilmington, DE). Quantitative real-time PCR analyses were performed in a StepOnePlus instrument (Applied Biosystems, Foster City, CA) in a 10-mL reaction volume containing 1 μL cDNA, 5 μL PerfeCta SYBR Green FastMix, Rox (2X) (Quanta BioSciences) and 0.5 μL of the primers listed in S2 Table. The thermal cycling conditions consisted of 20 s at 95˚C followed by 40 cycles of 3 s at 95˚C and 30 s at 60˚C. The amplification curves for the selected genes were parallel. The gene coding for eIF4E or ribosomal S16 was used as the endogenous control. Fold change (relative expression) was calculated using StepOne v2.1 and DataAssist v2.0 software (Applied Biosystems).

## Immunoblot analysis

Proteins from the parenchymal region of each mammary gland were extracted with RIPA buffer containing 10 mM Tris–HCl pH 7.5, 150 mM NaCl, 0.1% SDS, 1% sodium deoxycholate and 5 mM EDTA (Sigma), protease inhibitor cocktail and phosphatase inhibitor cocktail II (both from Sigma). Equal amounts of protein, determined by QPRO-BCA Kit Standard (Cyanagen, Bologna, Italy) according to the manufacturer's protocol, were subjected to SDS–PAGE. Fractionated proteins were transferred to nitrocellulose filters and the uploading of equal amounts of protein per sample was confirmed by staining the blot with Ponceau S (Sigma). The expression levels of phosphorylated S6 (pS6) and β-actin were determined according to the signals generated with their respective antibodies. Signals were generated with WESTAR Supernova (Cyanagen, Bologna, Italy) combined with Super Signal chemiluminescent substrate (Thermo Fisher Scientific), and detected in a Syngene analyzer (Cambridge, UK).

## Analysis of propagation rate

For propagation rate, cells from each treatment were seeded in 96-well cell-culture plates (Corning, Corning, NY) at a density of 10,000 cells/well and cultured in mammary medium composed of DMEM-F12 medium containing 5% FBS, hydrocortisone (0.5 mg/ml, Sigma), insulin (5 mg/ml, Sigma), gentamicin (50 mg/ml, Biological Industries), streptomycin and penicillin (100 mg/ml and 100 U/ml, respectively), hEGF (10 ng/ml, Merck, Darmstadt, Germany), hFGF2 (10 ng/ml, Merck), heparin (4 mg/ml, Merck), cholera toxin (10 ng/ml, Sigma) and B27 (4 ml stock/ml, Invitrogen, Carlsbad, CA) (10) for the indicated period.

## Organoid cultures

Partially digested pieces of parenchymal tissue were cultured on inserts (40 μm Falcon nylon cell strainers, Corning) in 6-well plates and supplemented with mammary medium [10]. Rapamycin (10 ng/mL) was administered to selected wells every other day. After a 3-week period, rapamycin-treated organoids were washed and all cultures were administered mammary medium for a week to avoid any further direct rapamycin effect. Medium was replaced every day. Eventually, the serum- and growth factor-containing medium was washed and milk protein gene expression was induced during 5 consecutive days of culture in DMEM/F12 medium containing (only) insulin (5 μg/mL), hydrocortisone (1 μg/mL) and prolactin (3 μg/mL). At the end of the culture period, organoids were frozen in liquid nitrogen and kept at -80˚C.

## Statistics

Unless otherwise indicated, t-test was performed for statistical analyses. Correlations were calculated using the Excel calculation formula.

## Results

### Establishment and calibration of rapamycin intramammary administration methodology

Three-month-old female calves were selected to analyze rapamycin's effect on mammary epithelial stem cell self-renewal and the potential consequences for development and production. The infrequently used intramammary drug-delivery method that engages the milk-mobilizing ductal system was preferred over systemic administration, to selectively confine rapamycin's effect to the mammary parenchyma (Fig 1A). To monitor vehicle mobility and penetration into the mammary epithelium, Evans blue (1%) was dissolved in the lipophilic solution and administered via the nipple to each of the four glands. Three hours later, the calf was sacrificed, the udder was removed and blue-stained parenchyma was revealed under the skin (Fig 1B). Note that in that short time interval, about half of the injected volume (~1 mL) was still located in the cisterna and did not penetrate the ductal system. Nevertheless, analysis of Carmine-red-counterstained mammary whole mounts revealed penetration of the blue stain into the ductal system and its translocation in the mammary epithelial terminal ductal lobuloalveolar units (TDLUs)—the ductal elongation region (Fig 1C).

To determine an effective concentration and administration interval, mTOR responsiveness to rapamycin administration was studied in parenchymal regions that were close to and far from the teat. Rapamycin was delivered to the parenchymatic region of two calves at two concentrations that were scaled up relative to the daily rapamycin administration to mice [[19] and S1A Fig]. The calves were sacrificed 24 h or 48 h later and the udders were removed. Rapamycin administration did not seem to have any effect on tissue morphology in the various parenchymatic regions tested (S1B–S1F Fig). S6 phosphorylation was analyzed in the mammary parenchyma by immunoblot and immunohistochemistry (Fig 1D–1L), respectively. These figures show that suppression of S6 phosphorylation following rapamycin administration at a level of 8.28 mg/2 mL in regions close to and far from the teat was maintained for at least 48 h. Note that complete repression of mTOR activity was avoided in these experiments to preserve the self-renewing capability of the epithelial stem cells.

### Rapamycin administration decreases mammary S6 phosphorylation and induces indicators of stem cell self-renewal

Based on the obtained information, we then investigated the effect of 3 weeks of skip-a-day rapamycin administration (8.32 mg/gland per day) on stem cell self-renewal (Fig 2). Here, a significant decrease of 53% in S6 phosphorylation was observed by immunoblot analysis, 48 h after termination of the experiment (Fig 2A and 2B), which was not associated with any detectable effect on β-casein synthesis (Fig 2C) or mammary gland morphology (Fig 2D). Immunohistochemical analyses localized the decreased pS6 expression exclusively to the mammary epithelial cells (Fig 2E) and confirmed comparable β-casein synthesis in vehicle- and rapamycin-treated glands (Fig 2F).

It was not technically feasible to perform flow cytometry analysis of rapamycin's effect on bovine stem cell self-renewal immediately after sacrificing the animals. Therefore, the parenchyma tissue of the individual mammary glands was partially digested into organoids [10] that were kept frozen at -80˚C. The organoids were later dissociated into a suspension of single cells for flow cytometry analysis using CD24 and CD49f antibodies (S1 Table and S2 Fig). A relatively low rate of cell survival and great diversity among samples were detected. Combined with the low (<1%) proportion of stem cells in the epithelial cell population, this hindered accurate detection of differences in stem cell number between rapamycin-treated glands and

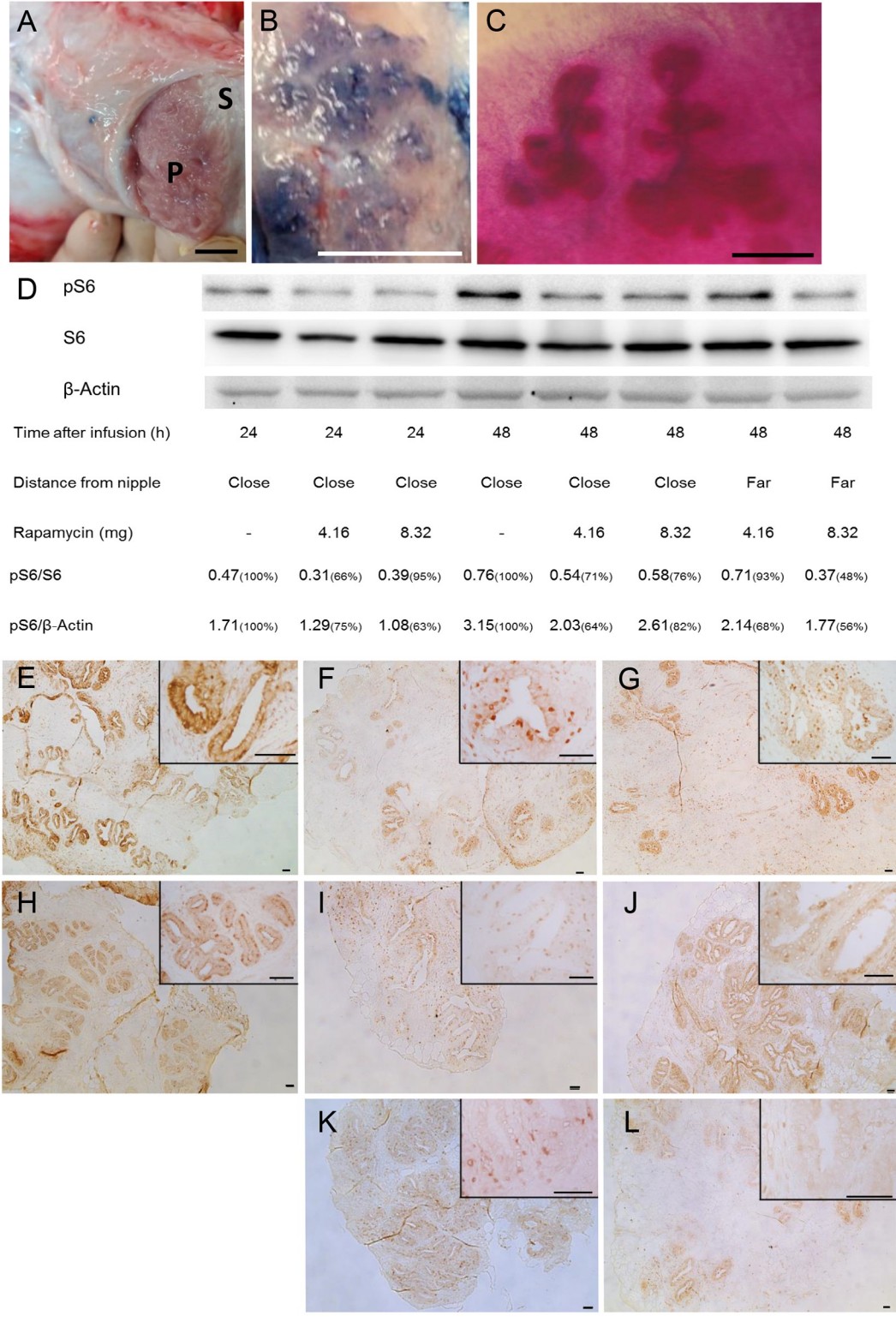

**Fig 1. Intramammary rapamycin administration reduces S6 phosphorylation in parenchymal cells.** A. Mammary gland morphology. P, parenchyma; S, stroma. Bar = 2 cm. B. Evans blue penetration of parenchyma, 2 h after administration. Bar = 1.25 cm. C. Mammary whole mount showing penetration of Evans blue into Carmine red-stained mammary epithelial structures. Bar = 1 mm. D. Immunoblot analysis showing a decrease in S6 phosphorylation (pS6) 24 and 48 h after rapamycin administration. E–G. Immunohistochemical analysis of S6 phosphorylation, 24 h after treatment, in regions close

to the nipple: E. Vehicle administration; F. rapamycin administration (4.16 mg); G. rapamycin administration (8.32 mg). H–
L. Immunohistochemical analysis of S6 phosphorylation, 48 h after treatment, in regions close to and far from the nipple: H.
Vehicle administration; I. Rapamycin administration (4.16 mg), close to nipple; J. rapamycin administration (8.32 mg), close
to nipple; K. Rapamycin administration (4.16 mg), far from nipple; L. Rapamycin administration (8.32 mg), far from nipple.
Bar = 50 μm.

their respective controls. Therefore, cell-propagation rate was analyzed in cultures from individual vehicle- and rapamycin-treated glands (Fig 3A). This method produces a clear and significant distinction of the bovine mammary epithelial stem cell population from the more differentiated ones [10]. It has also served as a reliable tool to analyze the effects of CR and rapamycin on mammary epithelial stem cell self-renewal in the endogenous mouse mammary gland and in bovine mammary implants, maintained in immunocompromised mice [19].

Here, a significant induction in cell number was initially detected in cultures from the vehicle-treated glands, but not in their rapamycin-treated counterparts. This probably represented propagation of partially differentiated epithelial cells with a relatively short survival rate, as evidenced by the consequent decrease in cell number. Indeed, by day 30 of culture, the number of cells from the vehicle-treated and rapamycin-treated glands equalized. In the long-term, higher propagation rate was observed for cultures from the rapamycin-treated gland, starting from 1.15-fold and 1.4-fold differences on days 37 and 52 of culture, respectively, and progressing to a 2.6-fold difference at the end of the culture period (Fig 3A inset).

There is no individual marker that specifically identifies bovine mammary stem cells [10, 11]. Nevertheless, a limited collection of genes that are highly expressed in mouse and human mammary stem cells has been analyzed in extracts of bovine mammary parenchyma [8, 24]. All of these markers were highly expressed in the rapamycin-treated glands, by 8–20% more than in their vehicle-treated counterparts (Fig 3B). In contrast, Stat5, which serves as a reliable reference for luminal progenitors [10, 11, 19, 24], was expressed at lower levels in the rapamycin-treated glands. Due to the high variability in expression among mammary glands, none of these results were significant at $P < 0.05$.

## Higher cell proliferation in rapamycin-treated glands depends on proximity to the growing site and studied cell layer

We then studied the consequences of rapamycin administration and induced stem cell self-renewal on mammary cell proliferation and milk protein expression. Front and rear mammary glands of three calves (six for each treatment) were infused with either vehicle or rapamycin for 3 weeks (S3 Fig). Five days after the last rapamycin administration, calves were subjected to daily estrogen and progesterone administration for 4 consecutive days to reproduce mammary cell proliferation during pregnancy [20]. At the end of the experimental period, animals were sacrificed, udders were removed and mammary parenchymal tissue pieces (~0.5 cm$^3$) were subjected to H&E staining and immunofluorescence analyses (Fig 4A and 4B). Ducts of various diameters were identified that penetrated the stroma and encompassed a multilayered luminal epithelium stained with CK18, lined by a single layer of basal myoepithelial cells stained with alpha smooth muscle actin (αSMA) (Fig 4B). There were no apparent differences in morphology between the vehicle- and rapamycin-treated glands. Tukey–Kramer (means comparison) statistical analysis ruled out any preferential effect on ductal cell number or diameter for an individual animal or gland (S3 Table). This enabled a more detailed characterization of the experimental system, by analyzing ducts in all vehicle-treated glands with a diameter of 51.35–191.81 μm (larger sizes did not enable distinguishing branching ducts from

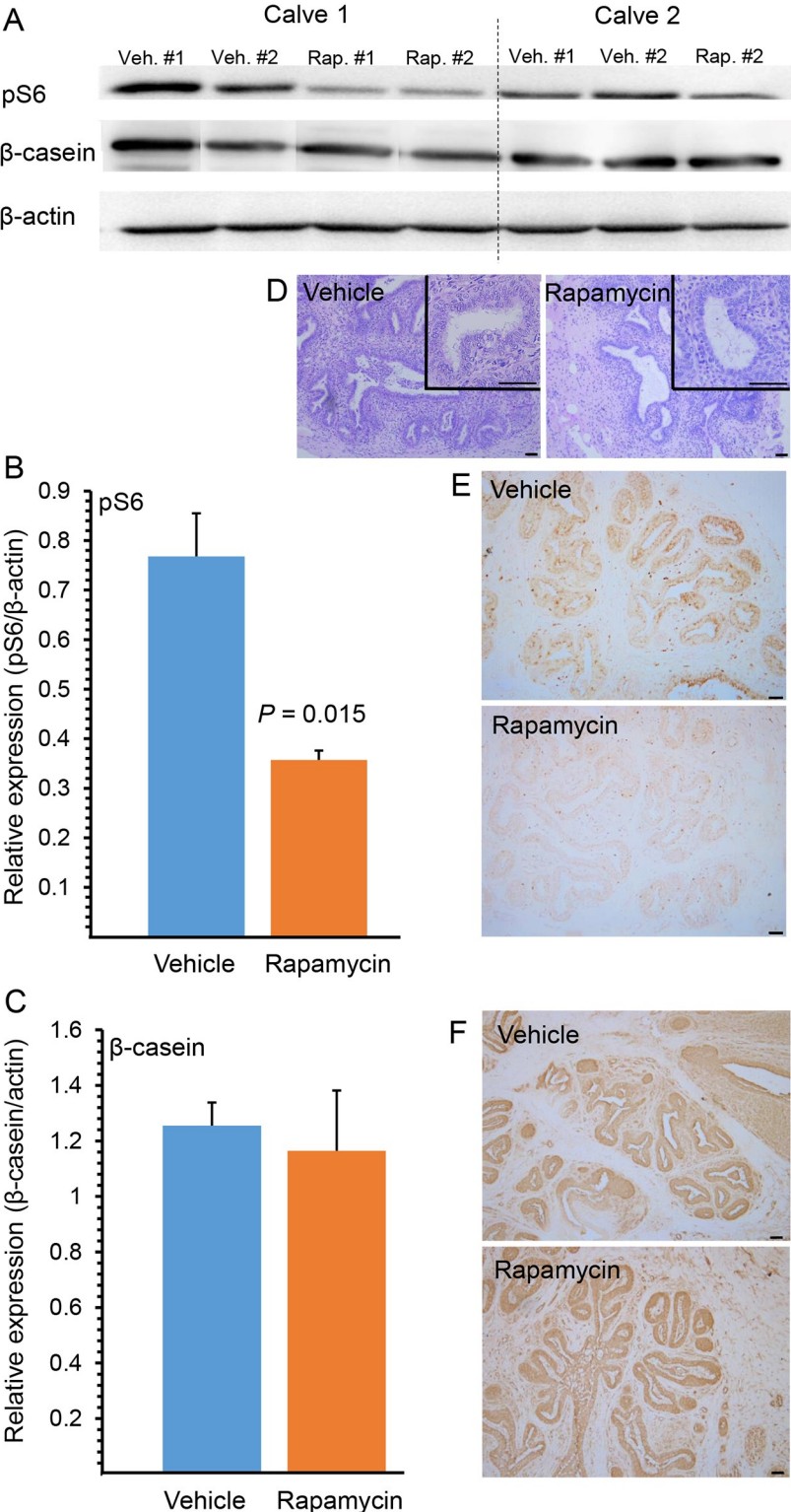

**Fig 2. Intramammary rapamycin administration for 3 weeks significantly reduces S6 phosphorylation in the mammary gland with no effect on mammary gland morphology or β-casein synthesis.** A. Immunoblot analysis of phosphorylated S6 (pS6) and β-casein. One of the nipples in calf #2 was inaccessible. B. Rapamycin administration causes a decrease in pS6 expression relative to β-actin expression (see A). C. Rapamycin administration does not affect β-casein expression relative to β-actin expression (see A). Bar graphs represent mean ± SEM of 3–4 replications. D.

H&E-stained mammary parenchymal sections demonstrating no effect of rapamycin on mammary morphology. Insets: higher magnifications. E. Representative immunohistochemical analysis demonstrating lower S6 phosphorylation in rapamycin-treated vs. vehicle-treated glands. Bar = 50 μm. F. Representative immunohistochemical analysis demonstrating no effect of rapamycin administration on β-casein expression in the mammary epithelial cells. Bar = 50 μm.

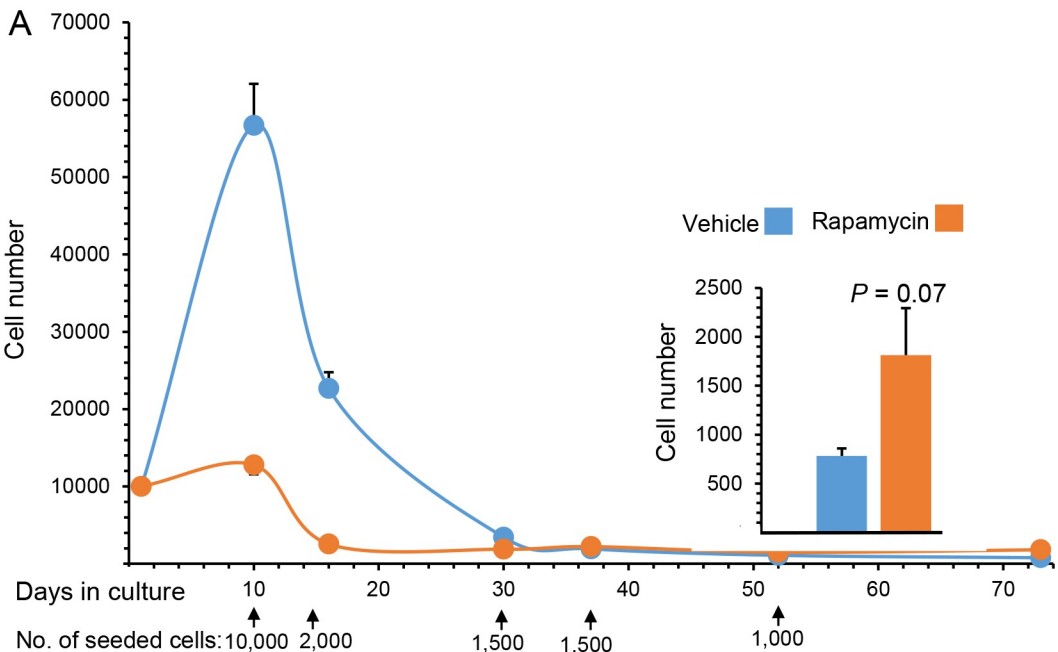

| | Relative gene expression (%) | |
| --- | --- | --- |
| | Vehicle | Rapamycin |
| Stat5 | 100±45 | 80±43 |
| Notch1 | 100±49 | 108±48 |
| Jagged1 | 100±54 | 110±75 |
| Delta1 | 100±34 | 115±44 |
| LGR4 | 100±36 | 120±69 |
| LGR5 | 100±15 | 110±20 |

**Fig 3. Intramammary rapamycin administration induces self-renewing cell properties.** A. Rapamycin administration induces propagation rate of cultured mammary epithelial cells. Dispersed cells isolated from individual vehicle- and rapamycin-treated mammary glands were seeded in 96-well plates and cultured in mammary medium. At the indicated time points (arrows), cells were trypsinized, counted and reseeded. Inset: cell number at the end of the culture period. Bars represent mean ± SEM. B. Relative gene expression analysis of luminal progenitor marker (Stat5) and stem cell markers (Notch1, Jagged1, Delta1, LGR4, LGR5). N = 4 for vehicle-treated and 3 for rapamycin-treated glands. Six wells of cells from each gland were analyzed.

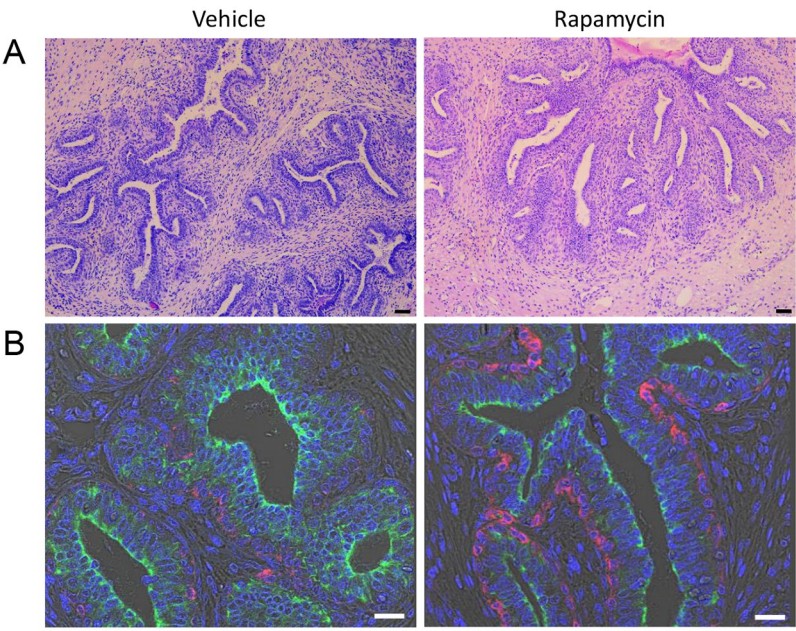

Vehicle | Rapamycin

A

B

C

| | Average ducts | | | Small ducts | | | Large ducts | | |
|---|---|---|---|---|---|---|---|---|---|
| Diameter/ perimeter | 111.3±2.3/ 349.7±7.4 | | | 85.4±1.87/ 268.21±5.8 | | | 125.7±2.3/ 394.7±7.2 | | |
| | Total | Basal | Luminal | Total | Basal | Luminal | Total | Basal | Luminal |
| Number of cells | 84.3±3.3 | 20.2±0.7 | 63.9±2.7 | 53.2±2.0 | 14.3±0.5 | 36.6±1.6 | 101.2±3.8 | 38.6±1.6 | 77.5±3.3 |
| % of total | 100 | 25.3±0.5 | 74.6±0.5 | 100 | 27.7±0.7 | 72.2±0.7 | 100 | 24.2±0.6 | 75.7±0.6 |
| PCNA+ cells | 28.7±1.1 | 8.5±0.3 | 20.2±0.2 | 21.7±0.92 | 6.5±0.3 | 15.2±0.7 | 32.4±1.4 | 9.6±0.4 | 22.8±1.2 |

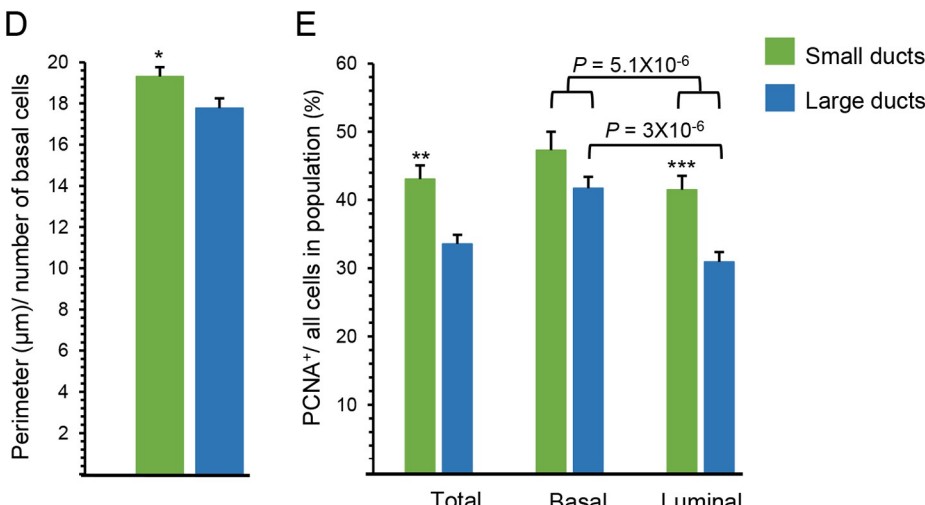

**Fig 4. Morphological and proliferative characteristics of 3-month-old vehicle-treated and rapamycin-treated bovine mammary gland after 4-day treatment with estrogen and progesterone.** A. Representative H&E staining of vehicle- and rapamycin-treated glands. Bar = 50 μm. B. Immunofluorescence analysis demonstrating multilayered luminal cells stained with CK18 (green) and a single layer of basal myoepithelial cells stained with αSMA (red). Bar = 20 μm. C-E. Mammary ductal cell characteristics of the vehicle-treated gland. C. Diameter size distinguishes small from large ducts. D. Density of basal cells in the ductal perimeter is higher in large vs. small

ducts. E. Number of PCNA$^+$ cells is higher in small vs. large ducts, mainly due to difference in luminal cell proliferation. Six independent microscopic fields (10X magnification) from each of the six glands treated with vehicle were analyzed. Overall, 127 ducts and 10,619 cells were counted. $^*P < 0.05$, $^{**}P < 0.01$, $^{***}P < 0.001$.

small independent ones; in smaller ducts, distinction of basal cells from stromal and immune cells was challenging). A cross section of an average-sized duct in a 3-month-old calf appeared to be comprised of 84 cells that interact to create a perimeter of ~350 μm (Fig 4C). Luminal cells represented most of the epithelial cell population, whereas the single-layered basal/myoe-pithelial cells made up only ~25% of the total cell number. Morphologically, these basal cells were significantly ($P < 0.05$) more densely packed in the perimeter of the large ducts (diameter >100 μm) compared to the small ones (diameter <100 μm) (Fig 4D).

To monitor cell proliferation, parenchymal sections from the individual mammary glands were stained with proliferating cell nuclear antigen (PCNA; Fig 5A and 5B). About 45% of the total number of cells in the vehicle-treated glands stained PCNA$^+$ (Fig 4C). The relative rate of basal cell proliferation was 1.26-fold higher ($P < 5.1 \times 10^{-6}$) than that of the luminal ones (Fig 4E).

Separate analyses were conducted for ducts with small and large diameters (Fig 4E), representing regions close to and far from the TDLUs, respectively [12, 19]. General proliferation rate was 28% higher in the small vs. large ducts, mainly due to 31% higher proliferation of luminal cells. Interestingly, in small ducts, no significant ($P < 0.05$) differences could be detected between the proliferation rates of basal and luminal cells, whereas in the large ducts, basal cell proliferation was 37% higher than that of luminal cells.

Rapamycin administration increased the total number of PCNA$^+$ cells by 20%, mainly due to a significant effect on cells composing the large ducts (Fig 5C). A more detailed analysis indicated that basal cell proliferation was induced by rapamycin in both small and large ducts, whereas a markedly higher rate of PCNA$^+$ luminal cells was detected only in the large ducts of the rapamycin-treated glands.

## Milk protein gene expression is induced in the rapamycin-treated glands, and is negatively correlated to endogenous milk protein expression potential

The latent effect of rapamycin administration on milk protein gene expression and synthesis was measured in the calves' mammary glands (Fig 6). The detectable expression of β-casein and β-lactoglobulin (BLG) in luminal mammary epithelial cells (Fig 6A) warranted more accurate gene and protein analyses of mammary productivity. Thus, expression of αS1-casein, β-casein, BLG and α-lactalbumin was analyzed by comparing the relevant gene expression in the rapamycin-treated glands of each animal to that in the vehicle-treated ones. In calf #2, rapamycin administration resulted in a latent 4.5- to 6-fold induction in the expression of all four milk protein genes tested, with no difference between the responses of casein and whey protein genes (Fig 6B–6E). This positive effect was confirmed by immunohistochemical analysis of β-casein, localized in the luminal epithelial cells (Fig 6G), and by immunoblot analysis (Fig 6F and S4 Fig). A modest effect of rapamycin on αS1-casein and BLG expression was observed in calf #3, but not in calf #1. Importantly, the various milk protein genes were expressed by mammary glands of calves #2 and #3 at levels of 0.4–12%, as compared to mammary glands of calf #1, which maintained much higher expression levels. Associating the average endogenous expression levels of the individual milk protein genes in the two vehicle-treated glands of each of the three calves to the inductive rapamycin effect (fold change), measured by comparing

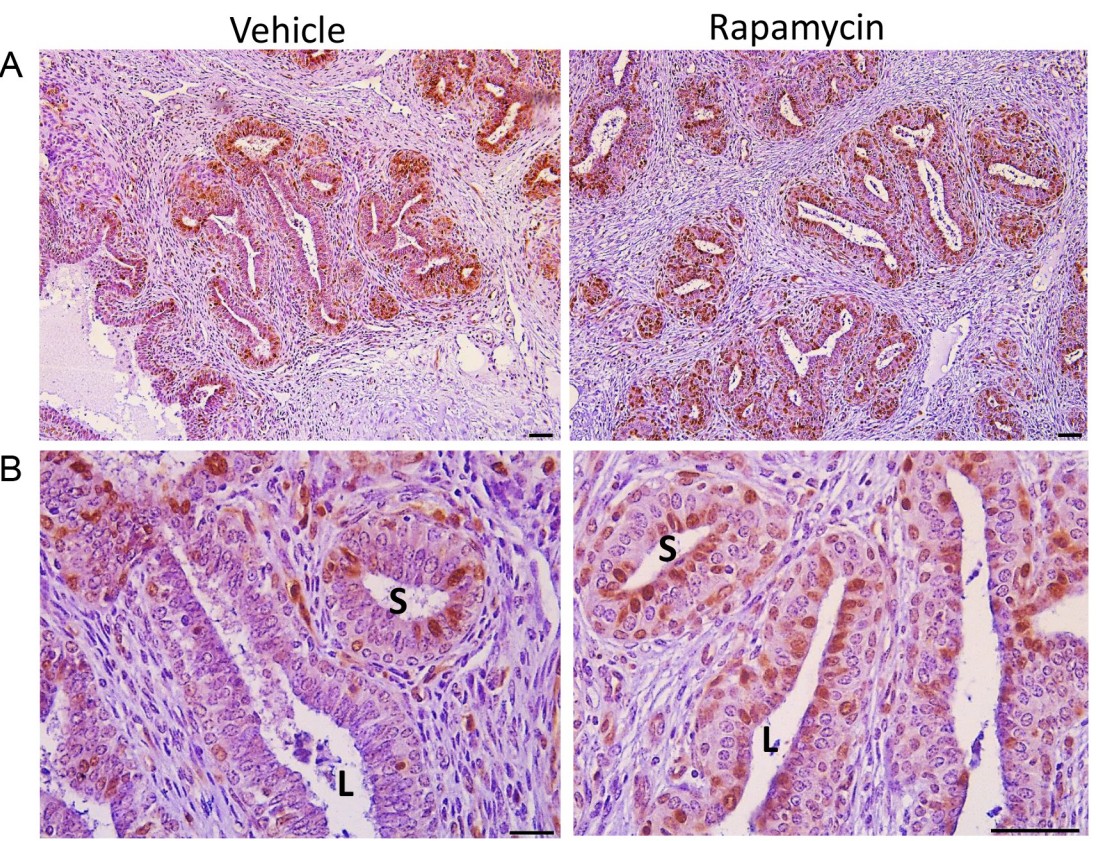

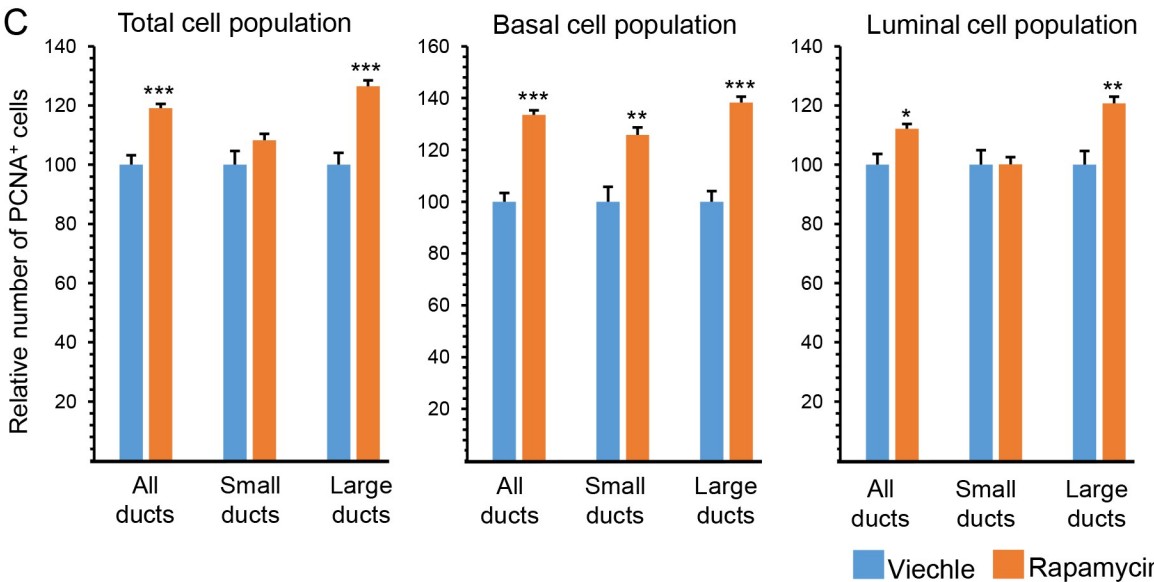

**Fig 5. Rapamycin administration results in latent induction of ductal cell proliferation in layer- and region-dependent manners.** Following estrogen and progesterone administration, mammary tissues were stained with PCNA and counterstained with hematoxylin. A, B. Lower (A) and higher (B) magnifications of PCNA-stained parenchymal cells. S, small ducts; L, large ducts. Bar = 50 μm. C. PCNA+ cells were analyzed in basal and luminal cell populations of the small and large ducts. Six independent microscopic fields (10X magnification) from each of the six glands treated with vehicle and six glands treated with rapamycin were analyzed. Overall, 127 ducts and 10,619 cells, and 122 ducts and 9999 cells, were analyzed in the vehicle-treated and rapamycin-treated glands, respectively. $^{**}P < 0.01$, $^{***}P < 0.001$.

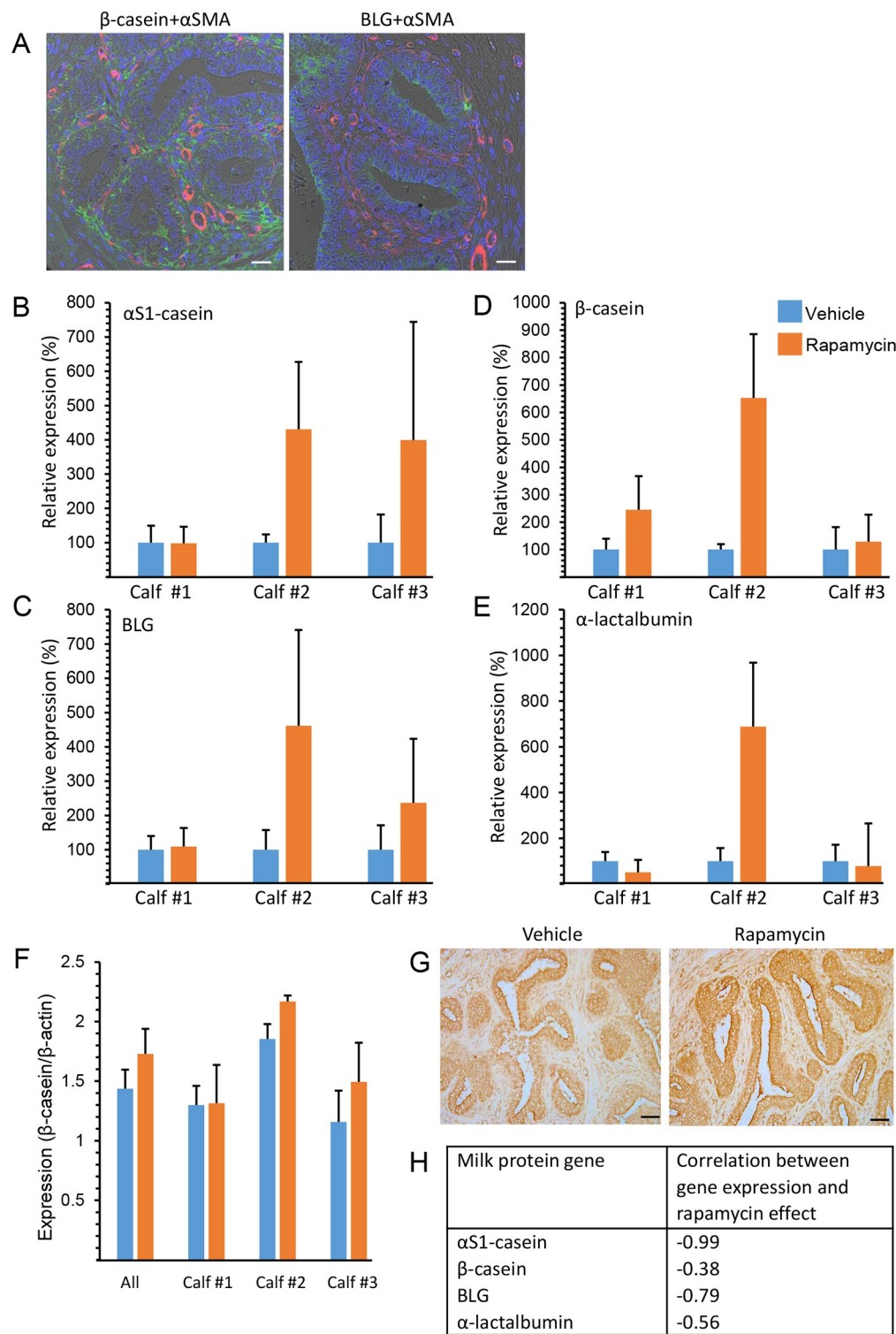

**Fig 6. Unequal effects of intramammary rapamycin administration on milk protein gene expression in individual calves.** A. Representative immunofluorescence analyses of milk protein expression by luminal cells in the mammary glands of 3-month-old calves. Bar = 20 μm. B–E. Effect of rapamycin on relative expression of milk protein genes is unequal among calves. Bars represent mean ± range of two glands in each analysis. F. Bar graph representing immunoblot analysis of β-casein expression relative to β-actin expression in the mammary glands of vehicle- and rapamycin-treated glands. Immunoblots are presented in S3 Fig. Bars represent mean ± range of two analyzed glands

for each calf except for "all", in which bars represent mean ± SEM of N = 6 (each) for vehicle- and rapamycin-treated glands. G. Immunohistochemical analysis supporting the gene-expression analysis and demonstrating higher β-casein expression in epithelial cells of the rapamycin-treated vs. vehicle-treated glands in calf #2. Bar = 50 μm. H. Negative correlation between endogenous levels of milk protein gene expression in the vehicle-treated glands of the individual calves and the respective positive rapamycin effect (fold change).

average gene expression in the two rapamycin-treated glands to their respective controls, presented a negative correlation of up to 99% for each gene (Fig 6H). Taken together, these results suggest that the positive rapamycin effect is limited by the endogenous expression potential of the mammary gland.

Mammary organoids preserve some, but not all interactions between epithelial cells and their surrounding stroma. By reproducing the procedure of the in-vivo study (except for steroid administration), we tested in organoid culture the latent animal-independent effect of 3-week rapamycin administration on milk protein gene expression in differentiating, lactogenic hormone-treated epithelial cells (S5 Fig). Significant latent induction of cytokeratin 18 (CK18) was observed by both gene expression and immunofluorescence analyses (S5A and S5B Fig), indicating a higher number of differentiated epithelial cells. However, in contrast to the in vivo finding, this was associated with a significant decrease in the expression of genes encoding all four milk proteins tested (S5C Fig). In fact, there was almost no β-casein synthesis or secretion in epithelial cells treated with rapamycin (S5B Fig).

## Discussion

Pharmacological studies in farm animals are often limited by the animals' relatively large size. The 3-month-old calf appears to have mammary parenchyma that is sufficiently well developed to scale up earlier findings in mice—associating decreased mTOR activity with induced epithelial stem cell self-renewal [19]—to the farm animal level. Using Evans blue marker dissolved in a lipophilic vehicle and counterstaining with Carmine red, intraductal delivery of material to the epithelium was established as an alternative to systemic administration. Targeting mammary epithelial cells around the ductal lumen, including those that are relatively far from the nipple, requires much less material compared to systemic administration and eliminates side effects resulting from the involvement of other organs. Indeed, the rapamycin delivered in this study to suppress mammary mTOR activity is an FDA-approved immunosuppressive drug, used to treat several types of cancer [25, 26] and recognized for its longevity-inducing effect [27]. However, long-term rapamycin administration disrupts mTOR complex 2 and subsequently results in hepatic insulin resistance leading to impairment of glucose homeostasis [28].

Intramammary rapamycin administration was calibrated according to S6 phosphorylation in the epithelial cells. A skip-a-day dose of 8.16 mg/gland per day was selected to satisfy three, somewhat contradictory, criteria: (i) providing a sufficient level of mTOR inhibition to induce putative paracrine secretion of stem cell self-renewal agent from as-yet unreported niche cells [[17, 19] and Kosenko et al. in preparation]; (ii) allowing sufficient mTOR activity in stem cells for self-renewal [18]; (iii) minimizing the risk of mammary infection during rapamycin administration. Indeed, by day 21 of the experiment, treated glands maintained about 47% of the mTOR activity in their vehicle-treated counterparts. This decrease was not associated with changes in mammary morphology or expression of the milk protein β-casein. In mice, inhibition of mTOR activity during the transition between pregnancy and lactation impairs terminal mammary differentiation, and β-casein synthesis is impaired in mammary HC11 cells [29]. A similar effect was seen with mTOR hyperactivity in tuberous sclerosis complex 1-knockout

(TSC1$^{-/-}$) mice, which could be reversed by rapamycin administration [30]. Hence, the amount and methodology chosen here for rapamycin administration did not have pathological consequences, but was able to induce stem cell self-renewal as depicted by the longer propagation rate of cultures originating from rapamycin-treated glands compared to their respective controls, and the collective increase in stem-cell marker expression. These results are in accordance with the rapamycin-induced stem cell self-renewal effect reported in the endogenous mouse mammary gland and in mature parenchymal bovine mammary implants maintained in the rapamycin-treated mouse mammary stroma [19]. They suggest a conserved mTOR-sensitive, age-independent effect among species.

Four days of estrogen and progesterone administration are sufficient for initial induction of bovine mammary cell proliferation [20]. Temporally constrained by ethical regulations, this steroid-administration procedure was reproduced here to follow the consequences of rapamycin administration during late pregnancy (Fig 7). Morphologically, the distance from the TDLU, depicted by increased ductal diameter, was negatively associated with proliferation rate as reflected by the significant decrease in luminal cell proliferation along the forming ducts. In the developing mouse mammary gland, most cell divisions occur in luminal progenitors and basal cells [31]. If this hierarchical definition applies to bovines, it suggests that actively proliferating basal/basal progenitor cells are more evenly distributed along the developing ducts compared to luminal progenitors, which are strongly accumulated at the TDLU sites [(see [5] for more information]. This finding is also in accordance with previous reports in humans, indicating that breast epithelial cells with a capacity for self-renewal, bipotency and clonal growth (i.e., stem cells) are derived from ducts, while active proliferating progenitors concentrate at the lobes [32]. The difference between the rates and locations of the luminal and basal cell proliferation may result from the higher proximity of the basal cells to external niche signals originating, for example, from ductal macrophages [33], as compared to their luminal counterparts.

Rapamycin administration resulted in latent induction of cell proliferation. Considering the unaffected β-casein expression at the end of rapamycin administration period, and the 9-day period separating the termination of rapamycin and the subsequent steroid administration, this positive effect probably does not represent short-term compensation, but rather reflects the consequences of the higher ratio of undifferentiated stem cells in the rapamycin-treated gland. Nevertheless, rapamycin induced cell proliferation selectively, discriminating large ducts with basal and luminal cell responses from small ducts in which only luminal cell proliferation is induced. This selectivity may be partly associated with the higher density of basal myoepithelial cells in the large ducts (i.e., far from the TDLUs) which may tightly control their proliferation.

The bovine mammary parenchyma seems to contain sufficiently differentiated epithelial cells at the age of 3 months to allow milk protein gene expression and protein synthesis. Indeed, expression of milk proteins was also observed in prepubertal buffalo mammary glands [34]. This suggests that the full repertoire of systemic and niche-dependent stimuli necessary for full differentiation of the epithelial stem cell toward a functional luminal cell already exists at this early developmental stage. Their activity in prepuberty dictates future productivity of the mammary gland [35, 36].

Rapamycin's positive effect on milk protein gene expression varied among calves; this was related to the gland's expression potential, determined by the relative number of epithelial cells in the parenchyma. Indeed, subjecting organoid cultures—which are insulated from the full repertoire of tissue and systemic effects—to rapamycin confirmed the latent induction of differentiated epithelial cell numbers observed in vivo. However, the lack of in-vivo oscillation in

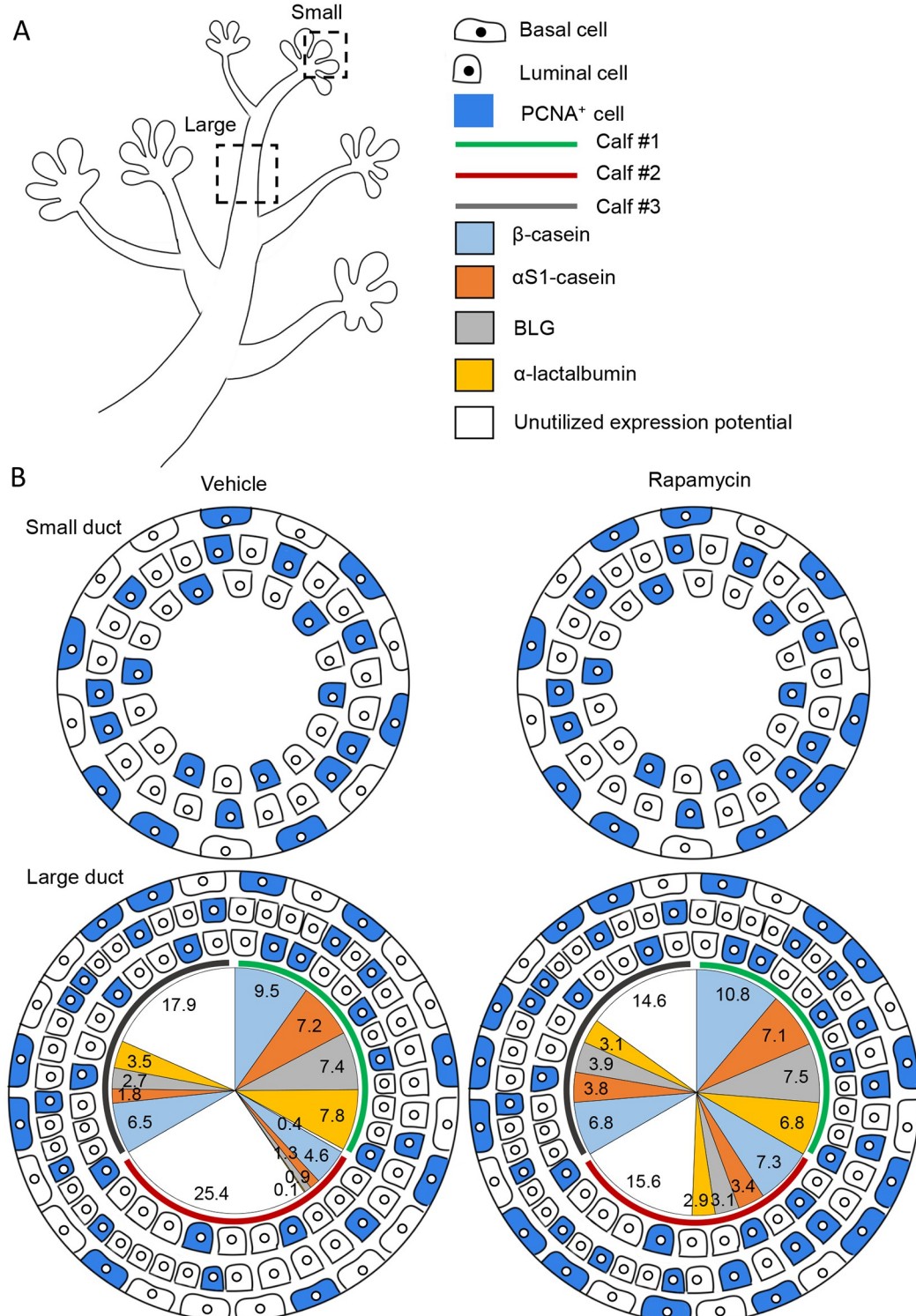

**Fig 7. Scheme summarizing the latent effects of intramammary rapamycin administration on cell proliferation and milk protein gene expression.** A. Representative small and large ducts in the developing mammary architecture are boxed. B. Basal and luminal cell proliferation in small and large ducts is presented in the outer three cell circles, where the relatively lower occupancy of basal cells in the small vs. large ducts can be visualized. Also demonstrated is rapamycin-induced cell proliferation in basal and luminal cells of the small ducts and only in luminal cells of the large ducts. The scheme is a precise representation of the relevant analyses. Milk protein gene expression in the mammary glands of the individual calves is

presented in pie chart format, inserted inside the circles showing the proliferation analysis of the large ducts. Presented are Log 2 (N+1) values of milk protein gene expression in the individual calves, calculated to accommodate and visualize the highly different values, some of which were lower than 1. The combined expression values of the four examined milk proteins in the mammary gland of rapamycin-treated calf #1 are highest and represent the maximum potential milk protein gene expression. Respectively, the pie chart demonstrates that rapamycin's inductive effect on milk protein gene expression is limited by the endogenous potential of the individual animal. Thus, almost no improvement in milk protein gene expression is detected in calf #1 with the highest level of endogenously expressed milk proteins. A stronger effect of rapamycin on milk protein gene expression is demonstrated calf #3, and even more so in calf #2 which expressed the lowest level of milk proteins (two orders of magnitude lower than calf #1), and maintained nonutilized expression potential.

rapamycin levels resulted in a continuous negative effect on milk protein expression, as was found in HC11 cultures [29].

## Conclusions

Intramammary administration of pharmaceuticals dissolved in lipophilic vehicle is an effective and efficient method to target mammary epithelial/myoepithelial cells in 3-month-old bovine mammary glands with a potential effect in the adult cow as well. A decrease of ~50% in mTOR activity by skip-a-day, 3-week intramammary rapamycin administration is sufficient to induce mammary epithelial stem cell self-renewal with no impairment of mammary morphology or milk protein production. The consequent improvements are induced cell proliferation and milk protein expression and synthesis in glands with relatively low milk protein gene expression. It reflects a genuine latent consequence of the rapamycin administration, followed by a higher number of self-renewed stem cells in the gland.

## Supporting information

**S1 Raw images. Complete blot for Figs 1D, 2B and 2C, and for Fig 6F.**
(TIF)

**S1 Table. List of antibodies used in this study.**
(DOCX)

**S2 Table. List of primers used in this study.**
(DOCX)

**S3 Table. Tukey–Kramer (means comparison) analysis of the average cell number/ duct and average ductal diameter in mammary glands treated with vehicle or rapamycin for 3 weeks followed by estrogen and progesterone.** The data indicate no significant effect for individual heifer or gland.
(DOCX)

**S1 Fig. H&E analysis demonstrating no effect after 48 h of intramammary rapamycin administration on mammary morphology.** A. Designation of the bovine mammary glands for intramammary vehicle and rapamycin administration. B–F. Morphology of parenchymal regions close to and far from the nipple is not affected by rapamycin administration.
Bar = 50 μm.
(TIF)

**S2 Fig. Flow cytometry analysis of dispersed CD24- and CD49f-labeled mammary cell population from 3-month-old calves.** Stem cells (MRU) were sorted from suspensions of single cells of individual vehicle- and rapamycin-treated glands.
(TIF)

**S3 Fig. Scheme demonstrating allocation of the mammary gland for vehicle and rapamycin administration.**
(TIF)

**S4 Fig. Immunoblot analysis of β-casein and β-actin expression in mammary glands treated with vehicle or rapamycin for 3 weeks followed by estrogen and progesterone administration.**
(TIF)

**S5 Fig. Latent inductive effect of rapamycin administration on CK18 expression in mammary organoids contrasts with its negative effect on milk protein expression.** Mammary organoids were cultured for 3 weeks in mammary medium supplemented, or not, with rapamycin. Rapamycin was omitted from the medium for an additional week, then the medium was changed to DMEM/F12 containing insulin, hydrocortisone and prolactin. A. Latent induction of CK18 relative expression by rapamycin administration. B. Immunofluorescence analysis demonstrating induced CK18 expression and decreased β-casein expression in rapamycin-treated organoids. Green: CK18 and β-casein. Red: αSMA. Bar = 20 μm. Inset: higher magnification. C. Negative effect of rapamycin on milk protein gene expression. Bars represent mean ± SEM of 4 replications. $^{*}P < 0.05$, $^{**}P < 0.01$, $^{***}P < 0.001$.
(TIF)

## Acknowledgments

We acknowledge Dr. Hen Hana Honig Felman for her contribution to the initial stages of the study and Dr. Tatiana Kisliouk for technical help.

## Author Contributions

**Conceptualization:** Itamar Barash.

**Investigation:** Anna Kosenko, Itamar Barash.

**Resources:** Shamay Jacoby, Maya Ross.

**Supervision:** Anna Kosenko, Tomer-Meir Salame, Maya Ross, Itamar Barash.

**Validation:** Anna Kosenko, Itamar Barash.

**Writing – original draft:** Itamar Barash.

**Writing – review & editing:** Itamar Barash.

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
