## [Decision Letter · Decision Letter 0]

7 Apr 2022

PONE-D-22-05135Intramammary rapamycin administration to calves induces epithelial stem cell self-renewal and latent cell proliferation and milk protein expressionPLOS ONE

Dear Dr. Barash,

Thank you for submitting your manuscript to PLOS ONE. After careful consideration, we feel that it has merit but does not fully meet PLOS ONE’s publication criteria as it currently stands. Therefore, we invite you to submit a revised version of the manuscript that addresses the points raised during the review process.

We look forward to receiving your revised manuscript.

Kind regards,

Sina Safayi, D.V.M., Ph.D.

Academic Editor

PLOS ONE

Journal Requirements:

Reviewers' comments:

Reviewer's Responses to Questions

**Comments to the Author**

1. Is the manuscript technically sound, and do the data support the conclusions?

Reviewer #1: Yes

Reviewer #2: Yes

2. Has the statistical analysis been performed appropriately and rigorously? 

Reviewer #1: Yes

Reviewer #2: Yes

3. Have the authors made all data underlying the findings in their manuscript fully available?

Reviewer #1: Yes

Reviewer #2: Yes

4. Is the manuscript presented in an intelligible fashion and written in standard English?

Reviewer #1: Yes

Reviewer #2: Yes

5. Review Comments to the Author

Reviewer #1: Overall this is a very well written manuscript that explores an interesting method for stimulating mammary stem cell renewal in a bovine model. This investigation is well-supported by previous literature and has considerable implications for the dairy industry. My reason for recommending minor revisions primarily pertains to the organization of the materials and methods. Particularly for the animal treatment section, it was difficult to understand how the animals were treated before they were sacrificed and the mammary glands removed. Providing a written version of the schema displayed in Figure S3 would help to clarify this. This manuscript also lacks a clear hypothesis. Please see my full review points below:

Abstract

• Well written, clear, no revisions necessary

Introduction

• Line 50 � “They also renew it between consecutive lactation periods”

o Is the “it” referring to the mammary epithelium or basal/luminal layer – please clarify

• Line 55 � add additional parentheses to (Reviewed by (4,9))

• Line 59-60 � “This rare population”

o Consider changing for clarity, bovine mammary cells are not rare but are seriously understudied

o Perhaps change to “This understudied population”

• Line 65 � add “the” between “that” and “mammary”

• Line 87 � change “to” to “for”

• Lacking hypothesis

Materials and Methods

• Lines 113-117 � unclear what was done, it is mentioned that a solution of sterile PBS containing

o Consider moving some of the information contained in lines 211-223 to the methods section

• 10% PEG and 10% Tween 80 was administered with or without the indicated supplements

o what supplements are you referring to?

o When were these supplements given?

o Overall, it is somewhat unclear what was done in the animals before they were sacrificed and the udder was removed – please clarify

• Line 131 � please add manufacturer information about cellSens software

• Line 192 � “mammary medium” – what is this?

o Also mentioned on lines 199-200

• Statistics � please indicate how correlations were analyzed

Results

• Lines 211-223 � most of this information is better suited in the materials and methods

• Lines 224-235 � same, consider moving most of this information into the materials and methods

o Perhaps just keep the information starting on line 229

• Lines 281-288 � consider moving this to the materials and methods

• Line 291 � change “difference” to “differences”

• Line 333 � remove “and that was”

• Line 339 � how were the correlations performed?

• Overall, very well written results section, figures are clear and explanations of what are presented are well done. Please consider moving some of the detailed methodology into the materials and methods section

Discussion

• Line 398 � add “which” between “progenitors” and “are”

• Very well written discussion. Usage of references was properly done. Implications of the data are clear

Conclusions

• Line 428 � remove ”the”

Supporting information

• Line 444 � change “Tuckey” to “Tukey”

Figure Legends

• Figures and figure legends were clear and easy to understand

• Figure 6 � whether in the caption, results, materials, and methods or all, a further explanation of how the correlations presented in Figure 6H were analyzed is needed

Reviewer #2: See attached.

6. PLOS authors have the option to publish the peer review history of their article (what does this mean?). If published, this will include your full peer review and any attached files.

Reviewer #1: **Yes: **Hunter Ford

Reviewer #2: **Yes: **Ratan K Choudhary

---

## [Author Response · Author response to Decision Letter 0]

26 Apr 2022

Reviewer #1

“Overall this is a very well written manuscript that explores an interesting method for stimulating mammary stem cell renewal in a bovine model. This investigation is well-supported by previous literature and has considerable implications for the dairy industry.”

Response: We thank the reviewer for expressing interest in the manuscript's topic and data.

 “My reason for recommending minor revisions primarily pertains to the organization of the materials and methods. Particularly for the animal treatment section, it was difficult to understand how the animals were treated before they were sacrificed and the mammary glands removed. Providing a written version of the schema displayed in Figure S3 would help to clarify this”.

Response: We thank the reviewer for this suggestion. The section related to animal treatment in Materials and Methods has been expanded to provide more details on how the experiments were conducted. It now includes a written version of the scheme displayed in Fig S3. 

“This manuscript also lacks a clear hypothesis”.

Response: The leading hypothesis has been added to the Introduction section. 

Abstract 

“Well written, clear, no revisions necessary”.

Response: Thank you 

Introduction

Line 50. “They also renew it between consecutive lactation periods. Is the “it” referring to the 

 mammary epithelium or basal/luminal layer – please clarify”.

Response: “It” refers to the mammary gland. The sentence has been revised accordingly.

Line 55. Add additional parentheses to (Reviewed by (4,9))”.

Response: Done

Line 59-60 “This rare population”. “Consider changing for clarity, bovine mammary cells are not rare but are seriously understudied”. Perhaps change to “This understudied population”

Response: The sentence has been revised as suggested.

Line 65 add “the” between “that” and “mammary”

Response: Done

Line 87 change “to” to “for””.

Response: Done

“Lacking hypothesis.”

Response: A clear hypothesis has been added toward the end of the Introduction section.

Materials and Methods

 Lines 113-117 "unclear what was done, it is mentioned that a solution of sterile PBS containing…Consider moving some of the information contained in lines 211-223 to the methods section.

Response: This section has been expanded to clarify the procedure. We carefully considered the reviewer’s suggestion to move part of, or the full section describing development, calibration, morphological verification and characterization of the intramammary methodology of rapamycin administration to the Materials and Methods. However, we feel that this section, including the associated figures, is an integral part of the research and supports the following experiments. Therefore, for the convenience of the reader and the integrity of the manuscript, we believe that it should remain in the Results section. Some of the data have already been included in Supporting Information. An example of integrating the development of experimental methodology in the Results section may be found in a previous study by Reichstein et al. “Conditional repression of STAT5 expression during lactation reveals its exclusive roles in mammary gland morphology, milk-protein gene expression, and neonate growth” Mol. Reprod. Dev. 78:585–596 (2011), where the section describing the development of the conditional Stat5 expression methodology, including the genetically modified mice, appears in the Results section, thereby providing a better understanding of the experiments that follow it.

 “10% PEG and 10% Tween 80 was administered with or without the indicated supplements. What supplements are you referring to? When were these supplements given? Overall, it is somewhat unclear what was done in the animals before they were sacrificed and the udder was removed – please clarify.

Response: A more detailed description of the methodology is now provided. 

Line 131. Please add manufacturer information about cellSens software.

Response: Done

 Line 192 “mammary medium” – what is this? “Also mentioned on lines 199-200”.

Response: The composition of the mammary medium and a reference is now provided. 

Statistics. Please indicate how correlations were analyzed.

Response: The technical aspect is now indicated in Materials and Methods. Further description is provided in the Results section in the paragraph referencing Fig 6H.

Results

Lines 211-223. Most of this information is better suited in the materials and methods.

Lines 224-235. “Same, consider moving most of this information into the materials and methods. Perhaps just keep the information starting on line 229”.

Response: Please refer to our explanation above as to why we prefer to keep this paragraph in the Results section. 

Lines 281-288. Consider moving this to the materials and methods.

Response: This paragraph describes the treatment of the animals and is essential, in our opinion, for understanding the following evaluation of the consequences of rapamycin administration. For a smooth narrative flow, we prefer to keep it as an introductory paragraph in the relevant subsection of the Results. 

Line 291, Change “difference” to “differences”.

Response: “There was no apparent difference between the morphology of the vehicle- and rapamycin-treated glands”. The “difference” is associated with the “morphology” and seems correct. Done 

Line 333. Remove “and that was”.

Response: Done

Line 339. How were the correlations performed?

Response: A more detailed description is now provided. The correlations were performed between two variables for each milk protein: (i) average milk protein gene expression level calculated for the two vehicle-treated (control) glands in each of the 3 analyzed calves; (ii) the effect of rapamycin on that particular gene's expression. The effect of rapamycin is the fold change obtained by comparing the average gene expression level in the two rapamycin-treated glands to its respective, vehicle-treated, control in the individual calf. 

Overall, very well written results section, figures are clear and explanations of what are presented are well done. Please consider moving some of the detailed methodology into the materials and methods section.

Response: Thank you. Please see our explanation above for wanting to keep some of the methodology in the Results section.

Discussion

Line 398. Add “which” between “progenitors” and “are”.

Response: Done

Very well written discussion. Usage of references was properly done. Implications of the data are clear.

Response: Thank you

Conclusions

Line 428. Remove” the”

Response: Done

Supporting information

 Line 444. Change “Tuckey” to “Tukey”.

Response: Done

Figure Legends

Figures and figure legends were clear and easy to understand.

Response: Thank you

Figure 6. Whether in the caption, results, materials, and methods or all, a further explanation of how the correlations presented in Figure 6H were analyzed is needed.

Response: The technical information is now provided in Materials and Methods. A more detailed explanation of how the correlations were performed is also provided (as mentioned above) in the Results section, in the paragraph referencing Fig 6H.

Reviewer #2

“…This is an exciting study and worthy of being considered for publication upon making suggestive changes”.

Response: Thank you

1. “Line 72-73: A study suggesting the role of xanthosine in increasing stem cell population is not limited to bovine only but has similar effects in the goat mammary gland (Choudhary et al., 2018; https://pubmed.ncbi.nlm.nih.gov/29804277/). Reference should be mentioned here”.

Response: A comment and the reference are now included in the text. 

2. “Line 183: Please elaborate on how protein concentration in the sample was determined to have an equal amount of protein per sample loaded for SDS-PAGE”.

Response: The information is now provided. 

3. Line 416-17: The presence of differentiated mammary epithelial cells in 3- months old calf is engaging. This information should be discussed a bit more here. This result is consistent with an earlier study where expression of milk protein genes was evident in the RNAseq study in the prepubertal buffalo mammary gland (Choudhary et al. 2019; https://pubmed.ncbi.nlm.nih.gov/30467802/). Please discuss this point here. 

Response: The presence of differentiated mammary epithelial cells in prepubertal calves was touched upon briefly so as not to divert the reader from the main focus of the manuscript. The reference has been included. 

4. Figure 2E: Please provide a representative picture of S6P staining of mammary tissue under the “Rapamycin” group. It is harder to visualize any cellular architecture of the tissue.

Response: The relatively low expression of pS6 caused by rapamycin administration indeed resulted in decreased DAB staining and visibility of the gland’s architecture. We have attempted to sharpen the figure to slightly improve visibility. We deliberately did not use hematoxylin counterstaining here so as not to mask the significant decrease in pS6 expression caused by rapamycin treatment. The architecture of the gland can be seen in Fig 2D. Please also consider the lower resolution of the figures in PDF format.

5. “Figure 5B: Please provide a representative picture of the under Rapamycin group showing a comparatively high number of PCNA-positive epithelial cells. It is challenging to visualize the increased cell proliferation rate in the rapamycin group”. 

Response: A more representative figure is now provided to demonstrate the higher number of PCNA-positive cells in the rapamycin-treated glands. Please note that the average difference between the total PCNA-stained nuclei in the control and rapamycin-treated glands is ~20% (Fig 5C).

Minor comments:

Line 638-639 and line 648-649: Please check if the information (duct no, cell no) is not the same in the legend of fig 4 and fig 5.

Response: In Fig 4C,D, only vehicle-treated “control” glands were analyzed for endogenous morphological and layer-associated aspects that characterize the 3-month-old calf mammary gland. Overall, 127 ducts and 10,619 cells were counted. In Fig 5, the rapamycin effect on latent induction of ductal cell proliferation was analyzed. For this, the same slides of the control mammary gland were used and the data were compared with the indicated and different number of 122 ducts and 9999 cells that were analyzed in the rapamycin-treated glands. 

We thank the reviewers for their comments and suggestions.

---

## [Decision Letter · Decision Letter 1]

23 May 2022

Intramammary rapamycin administration to calves induces epithelial stem cell self-renewal and latent cell proliferation and milk protein expression

PONE-D-22-05135R1

Dear Dr. Barash,

We’re pleased to inform you that your manuscript has been judged scientifically suitable for publication and will be formally accepted for publication once it meets all outstanding technical requirements.

Kind regards,

Sina Safayi, D.V.M., Ph.D.

Academic Editor

PLOS ONE

Additional Editor Comments (optional):

Reviewers' comments:

Reviewer's Responses to Questions

**Comments to the Author**

1. If the authors have adequately addressed your comments raised in a previous round of review and you feel that this manuscript is now acceptable for publication, you may indicate that here to bypass the “Comments to the Author” section, enter your conflict of interest statement in the “Confidential to Editor” section, and submit your "Accept" recommendation.

Reviewer #1: All comments have been addressed

Reviewer #2: All comments have been addressed

2. Is the manuscript technically sound, and do the data support the conclusions?

Reviewer #1: Yes

Reviewer #2: Yes

3. Has the statistical analysis been performed appropriately and rigorously? 

Reviewer #1: Yes

Reviewer #2: Yes

4. Have the authors made all data underlying the findings in their manuscript fully available?

Reviewer #1: Yes

Reviewer #2: Yes

5. Is the manuscript presented in an intelligible fashion and written in standard English?

Reviewer #1: Yes

Reviewer #2: Yes

6. Review Comments to the Author

Reviewer #1: Thank you for addressing all comments. I appreciate your explanation and justification for the information included in the materials and methods vs. the results section. After reading through the manuscript again, I understand your reasoning and agree with the decision to integrate some of the methodology into both sections.

Reviewer #2: Indeed this is good work performed by Barash group and has been doing a great work in the area of mammary gland and stem cell biology of ruminants, in particular bovines. Thank you for submitting your response and addressing our concern in revised version of manuscript. Wishing you and your team a great success!

7. PLOS authors have the option to publish the peer review history of their article (what does this mean?). If published, this will include your full peer review and any attached files.

Reviewer #1: **Yes: **Hunter R. Ford

Reviewer #2: **Yes: **Ratan K Choudhary

---

## [Editor Report · Acceptance letter]

30 May 2022

PONE-D-22-05135R1 

Intramammary rapamycin administration to calves induces epithelial stem cell self-renewal and latent cell proliferation and milk protein expression 

Dear Dr. Barash:

I'm pleased to inform you that your manuscript has been deemed suitable for publication in PLOS ONE. Congratulations! Your manuscript is now with our production department. 

Kind regards, 

on behalf of

Dr. Sina Safayi 

Academic Editor

PLOS ONE